# The effects of a synthetic epidermis spray on secondary intention wound healing in adult horses

**Paindaveine Charlotte C.**[1]❂*, **Bihin Benoit**[2], **Lepage Olivier M.**[1]❂

**1** Unité ICE-Groupe de Recherche en Médecine et Rééducation des Equidés de Sport (GREMERES), Centre for Equine Health, Ecole Nationale Vétérinaire de Lyon, VetAgro Sup, Université de Lyon, Lyon, France, **2** Namur Research Institute for Life Sciences (NARILIS), University of Namur, Namur, Belgium

❂ These authors contributed equally to this work.
* charlotte.paindaveine@vetagro-sup.fr

## Abstract

### Objectives

To evaluate secondary intention wound healing in the horse's limbs when treated with the synthetic epidermis spray (Novacika®, Cohesive S.A.S, France) or with a standard bandaging technique.

### Methods

Six Standardbred mares were included in the study. Four 2.5 x 2.5 cm full-thickness skin wounds were created on each thoracic limb. Two wounds were located on the dorsoproximal aspect of the cannon bone and the other two at the dorsoproximal aspect of the fetlock. Six hours after creation, wounds were randomly treated with synthetic epidermis spray or standard bandaging. The wounds were assessed every 4 days by gross visual assessment and using a 3D imaging camera. Analysis was performed with a 3D imaging application.

### Results

Out of 46 wounds, 22 showed exuberant granulation tissue and were part of the standard bandaging group. Whether the wounds were treated with synthetic epidermis spray or standard bandaging, the time for healing was the same.

### Conclusion

The synthetic epidermis spray studied in this model has allowed healing without the production of exuberant granulation tissue but did not reduce the median wound healing time compared to a standard bandaging technique. The synthetic epidermis spray is potentially an interesting alternative for the management of secondary intention wound healing of superficial and non-infected distal limb wounds in adult horses on economical and practical aspects. However, all statistical inference (p-values especially) must be interpreted with caution, given the size of the sample.

**Data Availability Statement:** All relevant data are within the manuscript and its Supporting Information files.

**Funding:** The study was funded by Cohesives SAS, Dijon, France. The funders did not play any role in the study design, data collection and analysis. The contract with the funders however specified that they have to accept the publication, which they did. The authors declare that there were no other conflicts of interest. https://www.societe.com/societe/cohesives-807497771.html https://novacika.com/ The funders had no role in study design, data collection and analysis, or preparation of the manuscript. However, the funders had the right to refuse publication of the study data and results.

**Competing interests:** The authors have declared that no competing interests exist.

## Introduction

Skin wounds are among the most common conditions affecting horses [1–3] and have a significant financial impact on the equine industry and caregivers [1,4,5]. A study in the United States revealed that skin wounds are the cause of 16% of euthanasia in adult horses and 23.9% in horses less than 6 months of age [6]. In a retrospective study including 500 injured horses and ponies, only 26% of the wounds in horses were successfully treated with primary closure [7]. Horses often live in highly contaminated environments and secondary intention wound healing is usually predominant [8–10]. However, wounds on the limbs can lead to complications and additional treatment is often required [2,11].

Exuberant granulation tissue (EGT) is the most common complication found on the limbs of a horse and should always be anticipated. The development of EGT is frequent due in particular to prolonged low-grade inflammation, which prevents wound contraction and proper epithelialization, leading to non-healing chronic wounds [4,12]. Wound complications impact the horse's athletic career and are the reason for prolonged and costly treatments. Therefore, alternative treatments are currently being investigated to improve secondary intention healing in horses [12].

Commonly, distal limb wounds are treated using nonadherent permeable dressing secured with conforming cotton gauze and held in place with a cohesive bandage.

In human medicine, multiple skin sprays have been studied and present numerous benefits such as the ease and short time of application, the possibility to treat large wound areas, and the homogeneous distribution of sprayed material [13]. The ease of application is particularly a major advantage for field practitionners. Acellular skin sprays generally consist of hydrogels that form a thin layer when sprayed over the wound and act as a protective dressing [13]. The synthetic epidermis spray (SES) used in this study is composed of ultraviolet polymerizable methacrylate monomers, comonomers, crosslinker and a photoinitiator. The methacrylate monomers are used as a base in multiple dressings in human medicine and seem to show excellent biocompatibility [14].

It was hypothesized that median wound healing time and prevalence of EGT formation would be lower on experimental wounds treated with a synthetic epidermis spray.

The study's objectives were to evaluate the median wound healing time and EGT prevalence on experimental open wounds in horses treated with the SES (Novacika®, Cohesive S.A.S., France) or with a standard bandaging technique.

## Material & methods

The study protocol was approved by the local animal care ethics committee and following the guidelines of the French Animal Ethics Committee (APAFIS #33883–2021111513453916). It was a prospective, randomized, controlled, cross-over experimental trial.

### Animals

Six healthy Standardbred adult mares of varying ages (range: 4 to 13 years old) and body weight (range: 425 to 560 kg), free of any scars or skin disease, were included in the study. Horses were considered healthy based on physical examination, hematology and biochemical panel results.

The horses arrived at the research facilities one week before the start of the study for acclimation. During the study they were housed in box stalls (10 m$^2$) and fed 10 kg of hay twice a day with water available *ad libitum*. General clinical examinations were performed daily throughout the study.

## Wound model

The protocol was based on a standardised wound model previously described [12,15]. The hair of both thoracic limbs was clipped from the dorsal aspect of the cannon (third metacarpal) bone the day before the surgical incision and wound creation.

On day 0, horses were sedated with detomidine chlorhydrate (0.01 mg/kg IV, Somnipron 10 mg/mL, Osalia, France) and butorphanol tartrate (0.02 mg/kg IV, Torbugesic VET 10 mg/mL, Zoetis, France). Additional sedation was administered as required for chemical restraint. Surgical sites were swabbed with a 0.9% NaCl solution prior to manipulation. Surgical scrub was not performed in order to mimic spontaneous trauma with natural contamination.

The skin was infiltrated approximately 2 cm above the surgical site with lidocaine hydrochloride 2% (Lurocaine 20 mg/mL, Vetoquinol, France) on each thoracic limb prior to surgical incision using a hemi-ring anaesthetic block technique. A template was used to standardise the creation of wound area. Using a size 15 blade and Metzenbaum scissors, a total of four 2.5 x 2.5 cm full-thickness skin wounds were created on each thoracic limb. Two wounds were located on the dorsoproximal aspect (one lateral and one medial) of the cannon bone and the other two at the dorsoproximal aspect (one lateral and one medial) of the fetlock. In the case of a hemorrhage with a continuous, pulsating flow of blood, a haemostatic forceps was applied for a period of 2 to 5 minutes. Each wound was numbered according to its location and was left uncovered for 6 hours after wound creation to mimic spontaneous trauma and field contamination.

## Treatments

The treatments were applied to the wounds in a randomised fashion with each horse acting as its own control. All thoracic limb wounds were subjected to one of the two treatments six hours after surgical wound creation. The horses were randomised into two groups (G1 and G2) by coin tossing. In the first group, the left thoracic limb was treated with treatment 1 (T1) and the right thoracic limb with treatment 2 (T2), while the second group received the opposite treatment. All wounds were cleaned with a 0.9% NaCl embedded swab before treatment application. T1 consisted of the application of the commercial SES composed of composed of ultraviolet polymerizable methacrylate monomers, comonomers, crosslinker and a photoinitiator. The spray was applied to the wound and polymerised using a 395 nm, 20 x 30 cm, 40 mW/cm$^2$ UV light (Cohesive S.A.S., France) for 60 seconds at a distance of approximately 30 cm (Fig 1). The same procedure was repeated to fix a second layer of the product. The control treatment (T2) consisted of a nonadherent permeable dressing (Aniplast SURGI, Génia, France) secured with conforming cotton gauze (Soffban Synthetic and Ouate Kistler, Alcyon, France) and held in place with a cohesive bandage (Vetrap and Tensoplast B, Alcyon, France) as common treatment of choice for superficial wounds in a field setting. The control treatment was repeated every 4 days until the end of the study to mimic field veterinary follow-up.

## Wound assessment

Wounds treated with T1 underwent daily gross visual assessment by the same evaluator (CP) for wound secretion, granulation tissue, contraction and epithelialization during a 60 days-period. T2-treated wounds were evaluated for the same criteria every 4 days at bandage change. EGT was defined as granulation tissue covering at least 3 out of 4 sides of the wound. Wound secretion, contraction and epithelialization were not statistically analysed but used to follow the wound healing.

A 3D digital image of each wound (treated with T1 and T2) was taken every 4 days using a 3D camera (3D LifViz®mini Quantificare, Biot, France), commonly used in human medicine

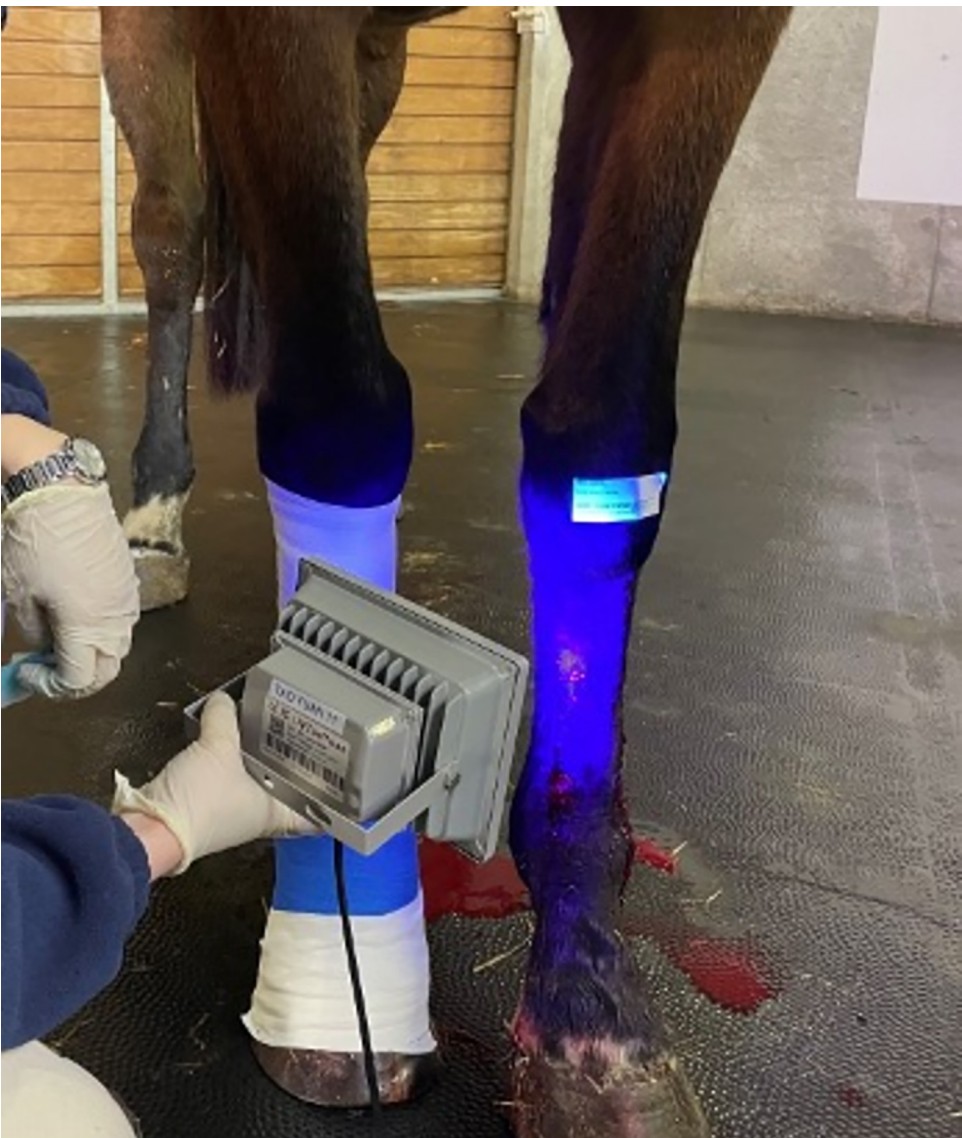

**Fig 1. Polymerization of the synthetic epidermis spray on a left thoracic limb using a 395 nm UV light for 60 seconds at a distance of approximately 30 cm.** The right thoracic limb has been treated with standard bandaging technique.

for maxillofacial surgery [16]. This camera uses two laser beams which indicate the optimal distance for image acquisition when the beams converge to a single point. The image was captured in 5 milliseconds and the images were then processed in dedicated 3D software (Lifeviz$^{TM}$App, Biot, France). After manual delineation of the wound contours by CP, the software automatically analysed the delineated area to obtain precise measurements of wound area ($cm^2$), volume ($cm^3$), height (mm) and depth (mm).

## Statistical analysis

In order to compare the wound healing kinetics in both treatments, generalized least square models with area, volume, height and depth as the dependent variables and time (in days), treatment (T1 vs T2), body part (cannon vs fetlock) and all interactions as the independent

variables were used. Because the outcomes kinetics (area, volume, height and depth) were not linear, we used restricted cubic splines to model the time effects [17,18]. When using restricted cubic splines, it is necessary to specify the number of knots used to approximate the non-linear relationship [17,18]. The choice of the knots number is a compromise between the model's ability to represent complex kinetics and the need to avoid overfitting. We thus used the Bayesian information criterion to determine the number of knots which are 8, 5, 8 and 6 to model area, volume, height and depth kinetics, respectively.

The use of generalized least square models was necessary because of the data structure. First, it was expected that two successive measurements on the same wound would be more correlated with each other than two measurements further apart in time. The model therefore included an auto-regressive (first-order) correlation structure that allowed for the time correlation. Second, variability in wound sizes was much greater when wounds were still large (during the first two weeks) than when wounds were nearly closed. To account for this heteroscedasticity, the model included a power variance function structure.

Observed values are showed alongside predicted values to describe how close the models fit to the data. Contrasts were used to compute differences between treatments with their 95% confidence intervals and P-values at four arbitrarily chosen timepoints: 0 days, 16 days, 32 days and 48 days.

A sensitivity analysis where the horse is the unit of analysis was performed and is presented in supplement.

In order to compare the EGT occurrence, wounds were first paired by horse and by zone, e.g., each pair comprises one wound treated with T2 and one wound treated with T1 for the same horse and on the same body part (but on the other limb). The wound pairs were then binned in four categories following the presence or absence of EGT. McNemar test was used to compare the EGT frequencies on paired data.

R 4.2.2 (The R Foundation for Statistical Computing, Austria, Vienna, 2022) was used for the statistical analysis with the following packages: *ggplot* (for graphical representations), *Hmisc*, *nlme* and *rms* (to handle generalized least square models and restricted cubic splines).

## Results

Two wounds were excluded from the study of wound healing kinetics because the horses moved during wound creation and their shape did not follow the original template. A total of 46 wounds instead of 48 were analyzed for a period of 60 days. Lameness was not observed in any horse during the study. Only 16 out of 46 wounds achieved a complete healing, defined by a complete epithelialization without scabs, by the end of the study.

The average evolution of wound area, volume, height and depth treated with T1 or T2 according to the anatomical zone is shown in Fig 2. Individual evolution of wound is shown in supporting information. The evolution of wound area treated with T1 or T2 was similar. Regarding cannon wounds, the difference between treatments (T2-T1) at 0, 16, 32 and 48 days were 0,8 cm$^2$ [95% confidence interval: -0.4cm$^2$ to 2.0 cm$^2$], 0.9 cm$^2$ [-0.2cm$^2$ to 2.0 cm$^2$], 0.5 cm$^2$ [-0.1 cm$^2$ to 1.1 cm$^2$] and 0.3 cm$^2$ [0.0 cm$^2$ to 0.6 cm$^2$], respectively (Table 1).

Fetlock wounds showed similar evolution (Fig 2 and Table 1).

The evolution of wound volume differed significantly between treatments for cannon and fetlock anatomical zone. After 16 days the T2 treated cannon and fetlock wounds volume were 0.44 cm$^3$ ([0.14 cm$^3$ to 0.74 cm$^3$], P = 0.004) and 0.24 cm$^3$ ([0.02 cm$^3$ to 0.45 cm$^3$], P = 0.03) higher than the T1 treated cannon wounds which represented relative increases in volume of 77% and 67%, respectively (Fig 2 and Table 1).

This volume difference is to be explained by differences in wound heights which showed a large peak around 16 days for T2-treated wounds but not for T1-treated wounds. The

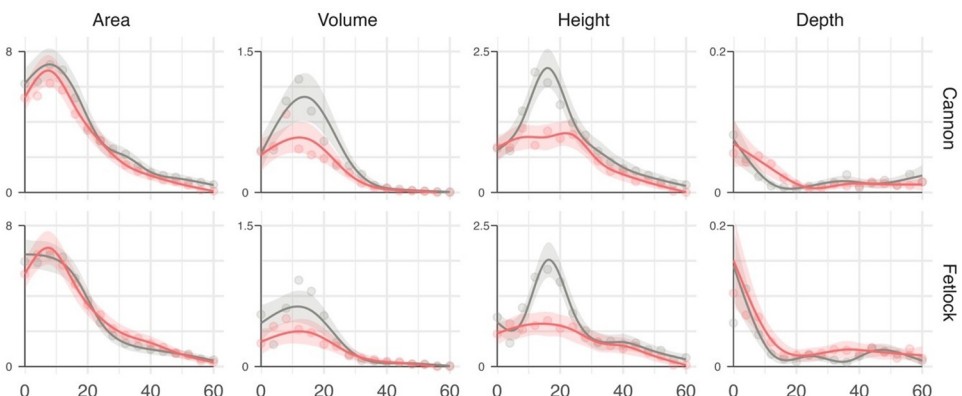

**Fig 2. Evolution of wound area, volume, height and depth.** Circles, solid lines and shaded area represent the observed means, the fitted values and the 95% confidence interval on the mean, respectively. Wounds treated with standard bandaging and SES are represented respectively in grey and red. The y-axis units are $cm^2$ for area, $cm^3$ for volume, cm for height and depth.

difference between treatment was +1.22 cm ([0.81 cm to 1.64 cm], P<0.0001), which represent a 124% increase, for cannon wounds and +1.13 cm ([0.75 cm to 1.51 cm], P<0.0001), which represent a 150% increase, for fetlock wounds (Fig 2 and Table 1).

The wounds depths showed a similar evolution for both treatments. The fetlock wounds were 0.07 cm ([0.01 cm to 0.13 cm], P = 0.03) deeper at start than cannon wounds.

Gross evaluation of secretion, contraction and epithelialization was only used as a follow-up for evolution of wound healing and not analyzed.

Wounds were assessed for EGT in 46 wounds for a period of 60 days (Fig 3). Wounds were paired by horse and by anatomical zone (1 to 4) and each pair consisted of one wound treated with T2 and one treated with T1. Out of 22 wounds treated with T1 and 24 treated with T2, 22 out of 24 treated with T2 showed EGT and none for T1-treated wounds. The EGT was far more frequent for T2-treated wounds than T1-treated wounds (20/22 vs 0/22 p<0.0001).

## Discussion

The results of this study suggest that the SES (T1) had a significant effect on the prevalence of EGT development in experimentally induced limb wounds in adult horses compared with a standard bandaging technique (T2). This was consistent with our hypothesis that the prevalence of EGT formation would be reduced in wounds treated with the SES compared to standard bandaging. However, a previous study showed that covering a wound with a non-occlusive dressing in a 3-layer bandage modulated the rate of wound contraction and promoted EGT [19]. It is therefore difficult to conclude that the absence of EGT on T1 treated wounds is related to the application of SES and/or the lack of covering bandage. A study comparing T1 treated wounds with non-treated wounds (naked wounds) could give further information on the matter.

It should be noted, that in our study, the EGT observed in the T2 group that did not receive the SES decreased without any trimming procedure during the 60-day period. Young oedematous granulation tissue protruding just above the wound margins has been described and generally does not require special treatment. The EGT may be characterized and differentiated by a firm and granular appearance [20]. In this study, the granulation tissue was considered exuberant. However, a histological study of this tissue at different moments of the healing process would be necessary to assess whether we are in the presence of a temporary increase in cell

**Table 1. Comparison of wound area, volume, height and depth at different timepoints.**

| | | | T1 | | T2 | | Difference between T1 and T2 | | |
|---|---|---|---|---|---|---|---|---|---|
| | | Day | Mean | SD | Mean | SD | T1-T2 | [95% CI] | P |
| Area | Cannon | 0 | 6.2 | 1.1 | 5.4 | 0.8 | 0.8 | [-0.4; 2.0] | 0.19 |
| | | 16 | 5.7 | 1.9 | 5,0 | 1.8 | 0.9 | [-0.2; 2.0] | 0.09 |
| | | 32 | 2.2 | 1.3 | 1.6 | 0.6 | 0.5 | [-0.1; 1.1] | 0.09 |
| | | 48 | 0.9 | 0.2 | 0.6 | 0.3 | 0.3 | [0.0; 0.6] | 0.09 |
| | Fetlock | 0 | 6.4 | 1.2 | 5.3 | 0.9 | 1.1 | [-0.1; 2.3] | 0.08 |
| | | 16 | 5.0 | 1.2 | 4.7 | 0.8 | 0.5 | [-0.6; 1.5] | 0.37 |
| | | 32 | 1.3 | 0.7 | 1.8 | 0.8 | -0.5 | [-1.0; 0.1] | 0.10 |
| | | 48 | 0.7 | 0.4 | 0.8 | 0.6 | 0.0 | [-0.4; 0.3] | 0.80 |
| Volume | Cannon | 0 | 0.41 | 0.20 | 0.39 | 0.28 | 0.01 | [-0.24; 0.26] | 0.92 |
| | | 16 | 1.16 | 0.70 | 0.57 | 0.60 | 0.44 | [0.14; 0.74] | 0.004 |
| | | 32 | 0.19 | 0.29 | 0.10 | 0.09 | 0.07 | [-0.01; 0.15] | 0.08 |
| | | 48 | 0.02 | 0.02 | 0.02 | 0.02 | 0.01 | [-0.01; 0.03] | 0.33 |
| | Fetlock | 0 | 0.50 | 0.42 | 0.25 | 0.16 | 0.20 | [-0.04; 0.43] | 0.10 |
| | | 16 | 0.80 | 0.30 | 0.36 | 0.13 | 0.24 | [0.02; 0.45] | 0.03 |
| | | 32 | 0.06 | 0.06 | 0.09 | 0.10 | -0.01 | [-0.07; 0.06] | 0.88 |
| | | 48 | 0.03 | 0.04 | 0.03 | 0.05 | 0.01 | [-0.02; 0.03] | 0.58 |
| Height | Cannon | 0 | 0.78 | 0.30 | 0.81 | 0.44 | -0.04 | [-0.35; 0.27] | 0.79 |
| | | 16 | 2.13 | 0.65 | 1.09 | 0.56 | 1.22 | [0.81; 1.64] | <0.0001 |
| | | 32 | 0.73 | 0.46 | 0.56 | 0.31 | 0.17 | [-0.09; 0.43] | 0.19 |
| | | 48 | 0.30 | 0.14 | 0.22 | 0.16 | 0.10 | [-0.03; 0.23] | 0.12 |
| | Fetlock | 0 | 0.88 | 0.55 | 0.58 | 0.26 | 0.20 | [-0.10; 0.50] | 0.20 |
| | | 16 | 1.72 | 0.32 | 0.81 | 0.26 | 1.13 | [0.75; 1.51] | <0.0001 |
| | | 32 | 0.42 | 0.20 | 0.44 | 0.26 | 0.02 | [-0.20; 0.25] | 0.84 |
| | | 48 | 0.30 | 0.31 | 0.25 | 0.24 | 0.08 | [-0.06; 0.23] | 0.27 |
| Depth | Cannon | 0 | 0.094 | 0.063 | 0.073 | 0.068 | 0.006 | [-0.039; 0.051] | 0.80 |
| | | 16 | 0.008 | 0.008 | 0.021 | 0.018 | -0.016 | [-0.027; -0.006] | 0.002 |
| | | 32 | 0.014 | 0.018 | 0.012 | 0.009 | 0.006 | [-0.002; 0.014] | 0.15 |
| | | 48 | 0.017 | 0.017 | 0.013 | 0.013 | 0.002 | [-0.007; 0.010] | 0.71 |
| | Fetlock | 0 | 0.083 | 0.080 | 0.104 | 0.046 | -0.008 | [-0.091; 0.075] | 0.85 |
| | | 16 | 0.011 | 0.008 | 0.016 | 0.011 | -0.013 | [-0.026; 0.001] | 0.06 |
| | | 32 | 0.009 | 0.009 | 0.024 | 0.026 | -0.014 | [-0.026; -0.003] | 0.02 |
| | | 48 | 0.022 | 0.031 | 0.020 | 0.016 | 0.004 | [-0.010; 0.017] | 0.60 |

Mean and SD (standard deviation) describe the observed values while the differences between wounds treated by T1 and by T2 (T1-T2), the 95% confidence interval (CI) and the P-values are computed from the generalized least square models.

volume, an increase in the number of cells followed by a decrease in their plasma volume or another interstitial phenomenon and therefore differentiate young oedematous granulation tissue or EGT. The authors chose to not perform biopsy to avoid any interference with the wound healing considering the small surface of the wound.

In the case of a distal limb wound in a horse, it is generally accepted that after cleaning the wound, a dressing should be applied and covered with a bandage to decrease contamination and oedema, absorb exudate, minimise movement and protect against more trauma [19]. It is recommended to change bandages every 3 to 5 days depending on the discharge and evolution of the wound healing [21]. To uniformize the method, the authors chose to change the bandages every 4 days.

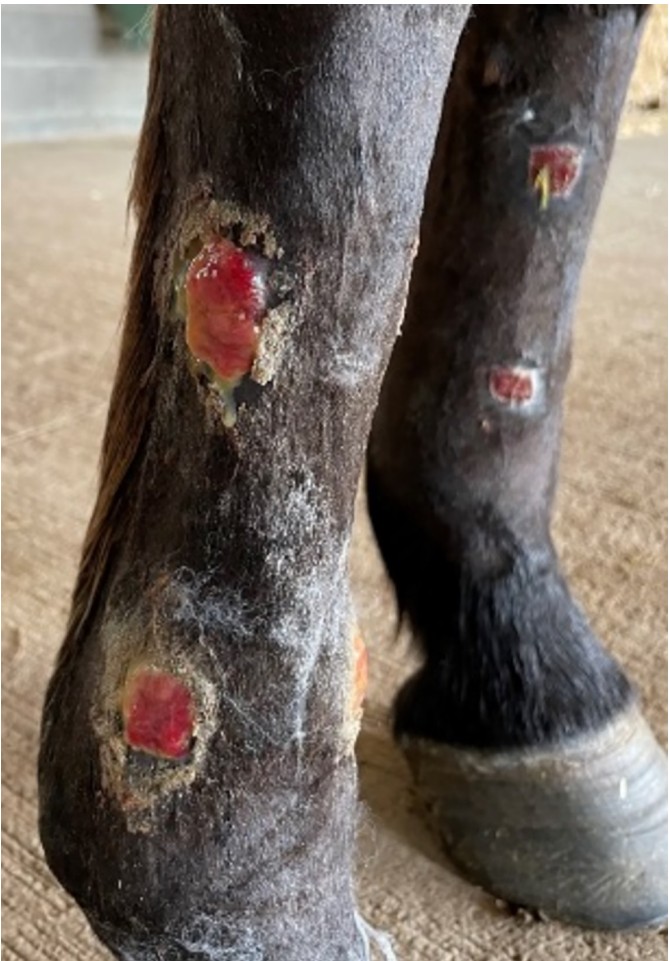

**Fig 3. Macroscopic aspect of healing thoracic limb wounds.** Hypergranulation and exsudative aspect of the lateral wounds is observed on the right limb treated with T2 compared to a flat and dry aspect on the medial left wounds treated with T1 in the same horse.

Contrary to our hypothesis that wounds treated with the synthetic epidermis spray would heal faster, no significant effect on the kinetics of the healing time was observed after day 32. The transient increase in wound volume that occurred in wounds treated with standard bandaging did not affect the healing time. This was supported by a previous report [19].

The SES was only applied once on day 0. No further application was made. The manufacturer describes a progressive resorption of its product. The application technique of the synthetic epidermis is easy and only requires the horse to be immobilised for approximately three minutes to allow polymerisation. The minimal material requirement also has a significant impact on the overall cost of treatment compared to standard bandaging changed every 4 days. In addition, this treatment could be an alternative for dangerous horses that do not tolerate bandages and could have an impact on the safety of veterinarians and owners. However, itching, external trauma or self-trauma to the wounds would not be prevented by this type of dressing.

There were several limitations associated to the present study, including the small number of horses. The first limitation relates to the animal species studied. The study protocol used and well described for horses is always carried out on a small number of animals [15]. The

horses were all adult and healthy and the results cannot be extrapolated to foals, elderly horses or individuals with a systemic disease. Although we deliberately left the wounds open for 6 hours before treatment to mimic normal contamination, for ethical reasons we had to create the wounds with a surgical technique that met the standards of cleanliness and sterilisation of the material. Similarly, our wounds only affect the skin tissue, whereas very often the deeper layers are also affected to varying degrees. The type of wounds studied here are similar to, naturally occurring, acute, poorly contaminated superficial degloving injuries and cannot be extrapolated to wounds presented at a more chronic stage, highly contaminated or infected, or affecting deeper structures such as bone or synovial cavities. The authors believe that deeper wounds would necessitate different care and therefore would not be treated in the first instance with SES. In addition, the study was conducted in winter excluding the presence of insects (e.g. flies).

## Conclusion

The synthetic epidermis spray (Novacika®, Cohesive S.A.S, France) studied in this wound model has allowed healing without the production of EGT but did not reduce the median wound healing time compared to a standard bandaging technique. The SES is potentially an interesting alternative for the management of secondary intention wound healing of superficial and non-infected distal limb wounds in adult horses because of its economical and practical aspects. However, all statistical inference (p-values especially) must be interpreted with caution, given the size of the sample.

## Supporting information

**S1 Table. Data of the wound assessment through the study period.**
(XLSX)

## Acknowledgments

The authors wish to acknowledge Maud Roux and Claire Forray for their technical help during the study.

## Author Contributions

**Conceptualization:** Paindaveine Charlotte C., Lepage Olivier M.

**Data curation:** Paindaveine Charlotte C.

**Formal analysis:** Bihin Benoit.

**Funding acquisition:** Lepage Olivier M.

**Methodology:** Paindaveine Charlotte C., Bihin Benoit, Lepage Olivier M.

**Writing – original draft:** Paindaveine Charlotte C., Bihin Benoit.

**Writing – review & editing:** Lepage Olivier M.

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
