## [Decision Letter · Decision Letter 0]

2 Jul 2023

PONE-D-23-16094The effects of a synthetic epidermis spray on secondary intention wound healing in adult horses.PLOS ONE

Dear Dr. Paindaveine,

Thank you for submitting your manuscript to PLOS ONE. After careful consideration, we feel that it has merit but does not fully meet PLOS ONE’s publication criteria as it currently stands. Therefore, we invite you to submit a revised version of the manuscript that addresses the points raised during the review process. Please submit your revised manuscript by Aug 16 2023 11:59PM. If you will need more time than this to complete your revisions, please reply to this message or contact the journal office at plosone@plos.org. Please include the following items when submitting your revised manuscript:A rebuttal letter that responds to each point raised by the academic editor and reviewer(s). You should upload this letter as a separate file labeled 'Response to Reviewers'.A marked-up copy of your manuscript that highlights changes made to the original version. You should upload this as a separate file labeled 'Revised Manuscript with Track Changes'.An unmarked version of your revised paper without tracked changes. You should upload this as a separate file labeled 'Manuscript'.

We look forward to receiving your revised manuscript.

Kind regards,

Carlos Alberto Antunes Viegas, DVM; MSc; PhD

Academic Editor

PLOS ONE

Journal Requirements:

"The study was funded by Cohesives SAS, Dijon, France. The funders did not play any role in the study design, data collection and analysis. The contract with the funders however specified that they have to accept the publication, which they did.

The authors declare that there were no other conflicts of interest.

https://www.societe.com/societe/cohesives-807497771.html

" ext-link-type="uri" xlink:type="simple">https://novacika.com/"

4. Please upload a new copy of Figure 6 as the detail is not clear. Please follow the link for more information: " ext-link-type="uri" xlink:type="simple">https://blogs.plos.org/plos/2019/06/looking-good-tips-for-creating-your-plos-figures-graphics/"
" ext-link-type="uri" xlink:type="simple">https://blogs.plos.org/plos/2019/06/looking-good-tips-for-creating-your-plos-figures-graphics/"

Reviewers' comments:

Reviewer's Responses to Questions

**Comments to the Author**

1. Is the manuscript technically sound, and do the data support the conclusions?

Reviewer #1: Partly

Reviewer #2: Partly

2. Has the statistical analysis been performed appropriately and rigorously? 

Reviewer #1: I Don't Know

Reviewer #2: I Don't Know

3. Have the authors made all data underlying the findings in their manuscript fully available?

Reviewer #1: Yes

Reviewer #2: Yes

4. Is the manuscript presented in an intelligible fashion and written in standard English?

Reviewer #1: Yes

Reviewer #2: No

5. Review Comments to the Author

Reviewer #1: Thanks for this manuscript.

I read this work with some attention and believe that this is a valid clinical research work. However, there are some obvious limitations to your study that you recognise yourself: Low numbers, non-standardised conditions for each animal, minor control of age range and animals acting as their own control.

I would like recommend some improvements if PlosOne are willing to take this on as a manuscript:

1. There is no introduction to the merits or reasoning for using the components in this wound dressing. There should some, although minimal intro of why this might be helpful: “Therefore, alternative treatments are currently being investigated to improve secondary intention healing in horses” is not enough. The reader needs to know why this type of dressing deserves a study of this nature.

I understand this work is sponsored by the wound dressing company. Fair enough but we need to know a bit more about the components of this material. Also, the type of lamp (size and power) are important aspects.

Other specific aspects:

Lines 83-85 This step introduces extremely high variability in the wound environment. Was there any type of microbiological understanding (ie. Culture) of the wound environment prior to treatment?

Lines 126-127 Area, height, depth and volume are related measurements and should be treated as such statistically. Also, Can you explain the difference between the measurements: depth and height? Figure 6 should detail this but seems to only be an illustration of the software interface.

Lines 136 “The numbers of knots used, respectively 8, 5, 8 and 6”, please explain this part of the text. I cannot relate to this.

Line 246: add itching, trauma and self trauma of the wounds as potential aspects that are not protected by this dressing

Lines 248-251 This conclusion is biased. If there is no advantage in healing of time of the wounds the advantages might be of economic and logistical order. This must be stated.

There are too many figures in the manuscript. I would recommend you reduce these to maximum 3.

Reviewer #2: PONE-D-23-16094

The effects of a synthetic epidermis spray on secondary intention wound healing in adult horses.

General comments

The authors report a prospective study on the effect of a synthetic spray on secondary intention wound healing in adult horses. The study presents the results of an original research that I believe has not been published elsewhere. Some details in the material and methods should be clarified to understand better the wound model (see specific comments). The paper’s conclusion seems appropriate, but some statements in the discussion are too strong for the results presented. I believe some major improvements could be achieved with the wording and how the figures and tables are presented. Most of the figures could be enhanced with better legend, annotations, and composition for better clarity for the reader. This paper could benefit from a more detailed and nuanced discussion. The research meets most applicable standards for the ethics of experimentation and research integrity. However, the study is funded by a company (Cohesive SAS), and it is not mentioned anywhere in the core of the paper that it is the company owning Novacika®. I believe the company’s name should be at least mentioned each time the product name is used in the paper.

With some improvements, this research has the potential to be a good fit for the journal.

Specific comments

Title

The title is appropriate, short, descriptive, and interesting for the reader.

Short title

The short title is the same as the long title.

Keywords

Keywords seem appropriate.

Abstract

L.15 (Novacika®)

L.21 remove “with”

L.23 evaluation?

L.29 “synthetic epidermis spray (Novacika®)”? o

Introduction

L34. Usually, in-text reference numbers should be inserted to the left or inside colons/semi-colons and placed outside or after full stops and commas. Please apply to the entire document.

L.40 “wounds on the limbs” or “limb wound” or “distal limb wounds”

L.42 “on the limbs”`

L.43 remove “in fact”

L.44 “inflammation, which”

L.46 “complications impact the horse’s…”

L.46 “horse’s athletic career”

L.49-53 I usually place the hypothesis before your objective as it describes how you plan to challenge your hypothesis, but it is up to debate…

L.51 “synthetic epidermis spray (Novacika®)”? Also, there are guidelines for citing trademarks as you did with detomidine: “detomidine chlorhydrate (0.01 mg/kg IV, Somnipron, 10 mg/mL, Osalia, France)”. Please be consistent throughout the paper when you add a trademark or a company/drug name. You could also use an abbreviation: SE or SES to simplify.

L.51 “or a standard bandaging…”

L.53 “hypothesized”

Having some data or comments to introduce the synthetic epidermis spray would be interesting. Are there any studies on similar products on other species, on humans? Description of the product, reported use, or effects? Also, maybe a comment on the other treatment is warranted. Why is it the comparative as opposed to no bandage at all?

Material and Methods

L.60 How do you justify the small number of horses in your study?

L.65 “acclimation”

L.71 Could you comment on the reason why you did not clip the hair on the same day as the surgery? As for our veterinary patients, clipping is recommended immediately before surgery.

L.75 Could you comment on the reason why you elected not to perform a surgical scrub before surgery? I assume it was to try to mimic a traumatic injury and/or to avoid skin flora disruption…it may be worth mentioning it.

L.77-78 Maybe specify where the regional bloc was performed

L.80-82 Based on your figures, it seems that wounds were more located on the dorsolateral and dorsomedial aspects of the limb, which does not fit your description in the text.

L.83 “hemostatic forceps”

See comments for Figure 1-3. I am convinced there is a better way of illustrating your wound model.

L.94-95 How did you randomize your horses? This is important to mention the method used.

L.102 How did you standardize the distance of UV exposition?

L.103-106 Why is your control to bandage wounds? Why not let the wounds heal without any bandages? You need to justify it or at least discuss it later on. As you mentioned in your discussion, bandage use promotes EGT, so this complication is to be expected down the line, so you need to justify your choice of control treatment. Otherwise, it will be difficult to compare the prevalence of EGT in both of your groups because if the wounds treated with the spray did not develop EGT, is that because they were not bandaged? Would naked wounds have developed EGT?

L.107 Why 4 days? Knowing what is reported in the literature about EGT and bandages, would a more frequent bandage change have been more appropriate to avoid EGT development?

L.114 “scoring system”

L.115 What were your criteria for each category of your contraction and epithelialization grading system? I understand it is a subjective grading system, but how could you visually assess contraction or epithelialization properly to classify it as mild, moderate, or severe?

L.117-118 What do you mean by “EGT aspect”? Do you mean “EGT was defined…”?

L.131-134 “…both treatments, a generalized…independent variable was used.”

Results

L.162-163 It would be interesting to know how long it took for the wounds to heal completely between both groups.

L.166 I am not able to see or download the supplementary material that is referenced in the paper

L164-199 This section is challenging to follow as the nomenclature does not match the one used in the table… P-values should be mentioned in the text for significant results

L183 Synthetic epidermis spray (Novacika,….) To simplify, you could use an abbreviation for it (SES) throughout the text.

L.191 Did you compare statistically treatments between locations? I do not see any p-value related to that

Discussion

L.206-208 I think it is an overstatement as the prevalence of EGT could be more related to the bandage application and the absence of it in the other group related to the lack of bandage… It is difficult to discriminate what decrease the occurrence of EGT between the spray and the absence of a bandage. Then the hypothesis could not be verified by the methods used in the study. There is probably a way to reformulate the conclusion.

L.210 Again, synthetic epidermis spray (Novacika….) or SES. Please apply to the rest of the manuscript.

L.216-217 This could be because it was just edematous granulation tissue that usually does not require any treatment. There is mention of this type of edema in the textbook Equine Wound Management by Dre Theoret et al. 3rd ed. (p378-379)

L.236 You need to justify the number of horses used. Maybe based on other studies using the same model?

L.239-240 In your MMs, it is not mentioned if the skin was surgically prepared before surgery. I assume it was to mimic an injury, but here it seems you did use a surgical technique to create the wounds… It is confusing; please clarify.

L.248 The conclusion is more appropriate to the study results than the beginning of the discussion.

The discussion seems a little bit short. Maybe you could talk about the pertinence of having a bandage group in your study vs. a non-bandaged group? Discuss the effect of the bandage (type, duration) on the production of EGT. Discuss the specific of the spray product, is it considered occlusive? Why do wounds not develop EGT under it? What is the evidence in other species that supports its use in horses? Why not biopsy the wounds throughout the healing to see if there is any difference histologically? Discuss the model used and the number of horses used. Could you have gone deeper into the tissues with your model to mimic classic trauma? Etc.

References

Reference 1 seems incomplete. Same for reference 4. There are missing pieces of information in many references, and the format does not fit the requirements of the journal… Please refer to the journal author guidelines to format your references…

Figures and Tables

Figure 1.

It does not represent your template well. You could try to have a picture of the template on a horse to help the reader understand how the template was used. If it is not possible, I would not keep the figure 1. Also, it seems that you used a permanent marker, but your protocol does not mention it. Was it a surgical

marker?

Also, the legend of a figure should always allow the reader to understand its significance without reading the text. Depending on whether or not you decide to keep the figure, you need to add more to your legend.

Figure 2.

This picture is blurry. Maybe a picture of a distal limb with all the fresh wounds would be more appropriate for the reader to understand the localization of the wounds, and you could potentially add your nomenclature (from Figure 3) to such a picture. Otherwise, I do not think that this picture is necessary.

Figure 3.

If you decide to keep this figure, maybe you could add little squares on your drawing to locate each wound. You could also combine that figure with a picture of your template, a picture of one wound, and a picture of each treatment. You add your protocol timeframe, giving you a nice study design figure to help any reader understand your protocol quickly.

Also, the legend does not correspond with what is shown in the figure…different nomenclature?

Figure 4.

I suggest adding the distance and the duration of the application in your legend.

Figure 5.

Figure 6.

The figure is extremely blurry. It may not be relevant. Maybe a figure showing the 3D camera in use and the results of the software?

Figure 7.

The figure is really blurry; you should use vector images for your graphics. It may be the pdf format that changes the resolution. It would be helpful to add some legends on the figure itself (colors, axis) and maybe consider a vertical organization with two columns of graphics (fetlock and metacarpus)

Table 1

Why are the groups named differently in the table (B, N)? Also, the table legend does not explain that, confusing the reader…

Figure 8

The background is blurry, so it is difficult to really appreciate the wounds on the left front limb…

6. PLOS authors have the option to publish the peer review history of their article (what does this mean?). If published, this will include your full peer review and any attached files.

Reviewer #1: No

Reviewer #2: No

---

## [Author Response · Author response to Decision Letter 0]

8 Sep 2023

Dear Reviewers,

Here are the answers to your comments and questions:

Reviewer #1 :

- Comment: Lines 83-85 This step introduces extremely high variability in the wound environment. Was there any type of microbiological understanding (ie. Culture) of the wound environment prior to treatment?

Answer: No there wasn’t. We believe that the environment contained multiple bacteria as it is on the field. We did not focus on the infection risk as it is rarely a problem in superficial wounds on horse’s limb. Culture is usually not performed on superficial wounds in clinical cases.

- Comment: Lines 126-127 Area, height, depth and volume are related measurements and should be treated as such statistically. Also, Can you explain the difference between the measurements: depth and height? Figure 6 should detail this but seems to only be an illustration of the software interface.

Answer: The wounds were analysed for area, volume, height and depth because wound healing on horse’s limb appears irregular with the granulation tissue being more prominent on a side. The height and depth values are giving information about the “landforms” of the granulation tissue. For example: the higher height would be where the granulation tissue was the more prominent and depth where there was poor granulation tissue.

- Comment: Lines 136 “The numbers of knots used, respectively 8, 5, 8 and 6”, please explain this part of the text. I cannot relate to this. 

Answer: This part of the text was adapted accordingly.

Reviewer #2:

- Comment: Usually, in-text reference numbers should be inserted to the left or inside colons/semi-colons and placed outside or after full stops and commas. Please apply to the entire document. 

Answer: I am not sure to understand this comment. I followed the journal requirement concerning the reference https://journals.plos.org/plosone/s/file?id=wjVg/PLOSOne_formatting_sample_main_body.pdf

- L.49-53 I usually place the hypothesis before your objective as it describes how you plan to challenge your hypothesis, but it is up to debate... 

Answer: we followed your advice

- Comment: L.60 How do you justify the small number of horses in your study? 

Answer: Above all, it's to respect the 3Rs (Replace, Reduce, and Refine) of animal research considered a systematic approach to animal experimentation that puts the well-being of the animal’s in front and by the fact that the experimental model we have used has already been validated many times in this large, expensive animal.

- Comment: L.71 Could you comment on the reason why you did not clip the hair on the same day as the surgery? As for our veterinary patients, clipping is recommended immediately before surgery. 

Answer: To facilitate the ease of the next day. It is true however that we should have done it the same day if we want to mimic natural occurring trauma.

- Comment: L.75 Could you comment on the reason why you elected not to perform a surgical scrub before surgery? I assume it was to try to mimic a traumatic injury and/or to avoid skin flora disruption...it may be worth mentioning it. 

Answer: indeed, we wanted to be as close as possible of the reality for natural occurring wound. We will add an explanation in the manuscript.

- Comment: L.102 How did you standardize the distance of UV exposition? 

Answer: Subjectively, it was the width of the manipulator's open hand (about 30 cm). A distance that could vary by a few centimetres if the horse moved its opposite limb.

- Comment: L.103-106 Why is your control to bandage wounds? Why not let the wounds heal without any bandages? You need to justify it or at least discuss it later on. As you mentioned in your discussion, bandage use promotes EGT, so this complication is to be expected down the line, so you need to justify your choice of control treatment. Otherwise, it will be difficult to compare the prevalence of EGT in both of your groups because if the wounds treated with the spray did not develop EGT, is that because they were not bandaged? Would naked wounds have developed EGT? 

Answer: we believe that another study should compare the product to naked wounds. However, in this study, we chose to compare to standard bandaging technique as most of the clinician cover wounds using this technique when facing superficial wounds. We explain this more clearly in the text.

- Comment: L.107 Why 4 days? Knowing what is reported in the literature about EGT and bandages, would a more frequent bandage change have been more appropriate to avoid EGT development? 

Answer: We chose to change it every 4 days as we believe veterinarian on the fields commonly change bandages every 3 to 5 days.

- Comment: L.115 What were your criteria for each category of your contraction and epithelialization grading system? I understand it is a subjective grading system, but how could you visually assess contraction or epithelialization properly to classify it as mild, moderate, or severe? 

Answer: As it is always the same assessor who inspects the wounds, he could more easily assess whether a smooth whitish-pink edge was developing around the periphery (epithelialization) and, or whether a reduction in the granulation surface was observed (contraction).

- Comment: L.162-163 It would be interesting to know how long it took for the wounds to heal completely between both groups. 

Answer: Regarding the area, only 16 wounds out of 46 achieve a complete healing (defined by a null wound area) during the follow-up. This is why we compared the wound kinetics instead of the time to complete healing.

- Comment: L164-199 P-values should be mentioned in the text for significant results

Answer: P-values were added in the text.

- Comment: L.191 Did you compare statistically treatments between locations? I do not see any p-value related to that 

Answer: We added the detail about the comparison between locations.

---

## [Decision Letter · Decision Letter 1]

20 Oct 2023

PONE-D-23-16094R1The effects of a synthetic epidermis spray on secondary intention wound healing in adult horses.PLOS ONE

Dear Dr. Paindaveine,

Thank you for submitting your manuscript to PLOS ONE. After careful consideration, we feel that it has merit but does not fully meet PLOS ONE’s publication criteria as it currently stands. Therefore, we invite you to submit a revised version of the manuscript that addresses the points raised during the review process.

We look forward to receiving your revised manuscript.

Kind regards,

Carlos Alberto Antunes Viegas, DVM; MSc; PhD

Academic Editor

PLOS ONE

Journal Requirements:

Reviewers' comments:

Reviewer's Responses to Questions

**Comments to the Author**

1. If the authors have adequately addressed your comments raised in a previous round of review and you feel that this manuscript is now acceptable for publication, you may indicate that here to bypass the “Comments to the Author” section, enter your conflict of interest statement in the “Confidential to Editor” section, and submit your "Accept" recommendation.

Reviewer #1: (No Response)

Reviewer #2: (No Response)

2. Is the manuscript technically sound, and do the data support the conclusions?

Reviewer #1: Yes

Reviewer #2: Yes

3. Has the statistical analysis been performed appropriately and rigorously? 

Reviewer #1: I Don't Know

Reviewer #2: I Don't Know

4. Have the authors made all data underlying the findings in their manuscript fully available?

Reviewer #1: Yes

Reviewer #2: Yes

5. Is the manuscript presented in an intelligible fashion and written in standard English?

Reviewer #1: Yes

Reviewer #2: No

6. Review Comments to the Author

Reviewer #1: Dear Authors,

Thanks for your reviews to the manuscript. I am overal happy with them.

Kind regards

Reviewer #2: L57 ‘, such as the ease and short time of application’

L57 ‘the short time of application’

L58 ‘areas, and the homogeneous’

L58-59 ‘The ease of application is particularly a major advantage for field practitioners.’

L61 ‘and act as a protective’

L64 ‘human’

L66-67 ‘It was hypothesized that median wound healing time and prevalence of EGT formation would be lower on experimental wounds treated with a synthetic epidermis spray.’

L69 ‘The study's objectives were to evaluate the median wound healing time and EGT prevalence on experimental open wounds in horses treated with the SES (Novika®, Cohesive S.A.S., France) or with a standard bandaging technique.’

L122 ‘as a common treatment of choice for superficial wounds in a field setting.’

L123 ‘field veterinary’

''- Comment: L.115 What were your criteria for each category of your contraction and epithelialization grading system? I understand it is a subjective grading system, but how could you visually assess contraction or epithelialization properly to classify it as mild, moderate, or severe?

Answer: As it is always the same assessor who inspects the wounds, he could more easily assess whether a smooth whitish-pink edge was developing around the periphery (epithelialization) and, or whether a reduction in the granulation surface was observed (contraction).''

Reviewer answer: That does not answer the question. For example, what is the difference between mild epithelialization, moderate epithelialization and severe epithelization? Same question for contraction. You used a subjective grading system without a reference atlas or measurements describing each category. That questions the pertinence of such assessment and its value in the paper. Later in the results section, you even mention that these assessments are not analyzed, so why do you bother explaining this subjective assessment? You could just mention that wounds were evaluated at each time point for the presence or absence of epithelization, contraction, EGT, etc.

L181 ‘defined by a complete epithelialization without scabs’

L185 Do you mean ‘supporting information’?

L185-225 Please use past tense when you report your results.

L221 Do you mean ‘Macroscopic aspect of healing thoracic limb wound’? These wounds are already granulating.

L222 ‘observed’

L224-225 I do not think this comment should be in the figure legend as it is not seen in the picture.

L227-232 The first paragraph of your discussion seems out of place. Maybe you could integrate it when you discuss the reported effect of bandages on the occurrence of EGT

L269 I already made a comment on the number of horses used. This has not be addressed. If you mention that one of your limitations is ‘the small number of horses’ you need to justify it by referring to other work/studies published with significant data with a similar model.

L283 ‘in this wound model’

L287-288 ‘…adult horses because of its economical and practical aspects.’?

References

There are still inconsistencies in the references...:

For example, AJVR is not a proper abbreviation…

Please use proper abbreviations for each journal (example of a database with journal abbreviations = https://images.webofknowledge.com/images/help/WOS/A_abrvjt.html)

Some references are incomplete (9, 12)

When you cite books, you usually have to specify the chapter with authors and pages. Do not cite an entire book.

7. PLOS authors have the option to publish the peer review history of their article (what does this mean?). If published, this will include your full peer review and any attached files.

Reviewer #1: No

Reviewer #2: No

---

## [Author Response · Author response to Decision Letter 1]

28 Nov 2023

Response to Reviewer 2,

Comment: L.115 What were your criteria for each category of your contraction and epithelialization grading system? I understand it is a subjective grading system, but how could you visually assess contraction or epithelialization properly to classify it as mild, moderate, or severe?

Answer: As it is always the same assessor who inspects the wounds, he could more easily assess whether a smooth whitish- pink edge was developing around the periphery (epithelialization) and, or whether a reduction in the granulation surface was observed (contraction).'' 

Reviewer answer: That does not answer the question. For example, what is the difference between mild epithelialization, moderate epithelialization and severe epithelization? Same question for contraction. You used a subjective grading system without a reference atlas or measurements describing each category. That questions the pertinence of such assessment and its value in the paper. Later in the results section, you even mention that these assessments are not analyzed, so why do you bother explaining this subjective assessment? You could just mention that wounds were evaluated at each time point for the presence or absence of epithelization, contraction, EGT, etc. 

Answer: please find a table in the file " Response to Reviewers" with the criteria for the grading system. However, as it was not analyzed, we will follow what you suggested.

Comment: L269 I already made a comment on the number of horses used. This has not be addressed. If you mention that one of your limitations is ‘the small number of horses’ you need to justify it by referring to other work/studies published with significant data with a similar model.

Answer: this protocol was used in a previous article (ref 15 in the manuscript) and showed that they could see a statistical difference with 6 horses. We added it on the text.

---

## [Decision Letter · Decision Letter 2]

12 Jan 2024

PONE-D-23-16094R2The effects of a synthetic epidermis spray on secondary intention wound healing in adult horses.PLOS ONE

Dear Dr. Paindaveine,

Thank you for submitting your manuscript to PLOS ONE. After careful consideration, we feel that it has merit but does not fully meet PLOS ONE’s publication criteria as it currently stands. Therefore, we invite you to submit a revised version of the manuscript that addresses the points raised during the review process.

If applicable, we recommend that you deposit your laboratory protocols in protocols.io to enhance the reproducibility of your results. Protocols.io assigns your protocol its own identifier (DOI) so that it can be cited independently in the future. For instructions see: https://journals.plos.org/plosone/s/submission-guidelines#loc-laboratory-protocols. Additionally, PLOS ONE offers an option for publishing peer-reviewed Lab Protocol articles, which describe protocols hosted on protocols.io. Read more information on sharing protocols at https://plos.org/protocols?utm_medium=editorial-emailutm_source=authorlettersutm_campaign=protocols.

We look forward to receiving your revised manuscript.

Kind regards,

Carlos Alberto Antunes Viegas, DVM; MSc; PhD

Academic Editor

PLOS ONE

Additional Editor Comments:

Dear author,

One of the initial reviewers asked pertinent questions about the statistical analysis of your work. In this sense, we requested support for the review in this particular aspect from a statistics specialist. This recommends a major revision in order to redo some aspects of the statistical treatment.

Best regards

Reviewers' comments:

Reviewer's Responses to Questions

**Comments to the Author**

1. If the authors have adequately addressed your comments raised in a previous round of review and you feel that this manuscript is now acceptable for publication, you may indicate that here to bypass the “Comments to the Author” section, enter your conflict of interest statement in the “Confidential to Editor” section, and submit your "Accept" recommendation.

Reviewer #2: All comments have been addressed

Reviewer #3: (No Response)

2. Is the manuscript technically sound, and do the data support the conclusions?

Reviewer #2: Yes

Reviewer #3: No

3. Has the statistical analysis been performed appropriately and rigorously? 

Reviewer #2: I Don't Know

Reviewer #3: No

4. Have the authors made all data underlying the findings in their manuscript fully available?

Reviewer #2: Yes

Reviewer #3: No

5. Is the manuscript presented in an intelligible fashion and written in standard English?

Reviewer #2: Yes

Reviewer #3: Yes

6. Review Comments to the Author

Reviewer #2: Dear authors,

Thank you for your work on the manuscript ! I am satisfied with its actual form.

Kind regards

Reviewer #3: The authors set out to evaluate the secondary intention wound healing in the horse's limbs when treated with the synthetic epidermis spray or with a standard bandaging technique. The design and analysis appears to be simple using the generalized least square models with area, volume, height and depth as the dependent variables and time (in days), treatment (T1 vs T2), body part (cannon vs fetlock) and all interactions as the independent variables . The interpretation, as noted below, is not simple. Also the investigators claim that six horses are adequate for the sample size as noted from previous publications. However, the sample size is not discussed and certainly not justified with respect to any of the variables in this presentation.

Simply stated the overall result is that whether the wounds were treated with synthetic epidermis spray or standard bandaging, the time for healing was the same. This is not obvious in light of the time and location effect. The presentation of the results are seen primarily in Table 2. It appears like a multivariate issue with the major variables of interest being time, treatment and location. There are no gross p-values where the authors state in the overall experiment whether there is an independent time, treatment or location effect adjusted for the others. The attempt at explaining the interactions may be reasonable. Table 2 is a summary univariate analysis for the dependent variables in five tables. The organization of the analysis is confusing. There are many analyses and many p-values, resulting in possible multiple comparisons and no adjustment for such.

Also, the unit of analysis is the wound (it appears). The horse may also be a unit of analysis which gets into the issue of intraclass correlation of the treatments across these units. The variance of the estimates could be

inflated in this situation.

The unit of analysis issue and multiple tests could possibly be addressed by the authors already as seen on line 157 in which they state that the model therefore included an auto-regressive (first-order) correlation structure that allowed for this or is this addressing the spatial (location) and time correlation? All of this has to be better clarified by the investigators.

7. PLOS authors have the option to publish the peer review history of their article (what does this mean?). If published, this will include your full peer review and any attached files.

Reviewer #2: No

Reviewer #3: No

---

## [Author Response · Author response to Decision Letter 2]

1 Feb 2024

Response to Reviewer #3: 

Comment: The authors set out to evaluate the secondary intention wound healing in the horse's limbs when treated with the synthetic epidermis spray or with a standard bandaging technique. The design and analysis appears to be simple using the generalized least square models with area, volume, height and depth as the dependent variables and time (in days), treatment (T1 vs T2), body part (cannon vs fetlock) and all interactions as the independent variables. The interpretation, as noted below, is not simple. Also the investigators claim that six horses are adequate for the sample size as noted from previous publications. However, the sample size is not discussed and certainly not justified with respect to any of the variables in this presentation.

Simply stated the overall result is that whether the wounds were treated with synthetic epidermis spray or standard bandaging, the time for healing was the same. This is not obvious in light of the time and location effect. The presentation of the results are seen primarily in Table 2. It appears like a multivariate issue with the major variables of interest being time, treatment and location. There are no gross p-values where the authors state in the overall experiment whether there is an independent time, treatment or location effect adjusted for the others. The attempt at explaining the interactions may be reasonable. Table 2 is a summary univariate analysis for the dependent variables in five tables. The organization of the analysis is confusing. There are many analyses and many p-values, resulting in possible multiple comparisons and no adjustment for such.

Also, the unit of analysis is the wound (it appears). The horse may also be a unit of analysis which gets into the issue of intraclass correlation of the treatments across these units. The variance of the estimates could be inflated in this situation.

The unit of analysis issue and multiple tests could possibly be addressed by the authors already as seen on line 157 in which they state that the model therefore included an auto-regressive (first-order) correlation structure that allowed for this or is this addressing the spatial (location) and time correlation? All of this has to be better clarified by the investigators.

Answer: Thank you for your remarks. 

1. We clarified the generalized least square model concerning the auto-regressive correlation structure. 

2. We included an additional table presenting a simplified version of the analysis, which can be considered a sensitivity analysis, with the horse as the unit of analysis. We computed and averaged the area under the curve for each wound kinetics, grouped by horse and treatment (refer to the table below). The conclusions mirror those of the complex model: the height is observed to decrease with SES bandaging, while the area and volume remain unaffected. 

Correction for multiple comparisons does not seem necessary, as observations on the various outcomes at different times are highly dependent.

Concerning the sample size, it was predefined based on material constraints. A posteriori, this sample size proved adequate to highlight the treatment effect on wound height evolution and convincingly demonstrate that the healing duration is not strongly influenced by the type of bandage.

 Area under the curve 95% CI 

Horse Outcome B N Difference Mean lower limit upper limit P

A Area 177 234 -57 8 -45 61 0,70

C 166 202 -36 

H 172 141 31 

I 218 172 46 

V 183 191 -8 

Y 233 158 75 

A Depth 1,9 2,0 -0,1 -0,3 -0,6 0,0 0,07

C 1,4 1,9 -0,5 

H 1,2 1,9 -0,7 

I 1,6 1,6 0,0 

V 1,3 1,5 -0,2 

Y 2,2 2,3 -0,1 

A Height 49 51 -2 16 1 31 0,04

C 48 42 6 

H 44 24 20 

I 55 29 26 

V 50 40 11 

Y 60 24 37 

A Volume 16 21 -5 8 -3 19 0,13

C 17 17 0 

H 17 7 10 

I 27 13 14 

V 19 16 3 

Y 31 7 24

---

## [Decision Letter · Decision Letter 3]

7 Feb 2024

PONE-D-23-16094R3The effects of a synthetic epidermis spray on secondary intention wound healing in adult horses.PLOS ONE

Dear Dr. Paindaveine,

Thank you for submitting your manuscript to PLOS ONE. After careful consideration, we feel that it has merit but does not fully meet PLOS ONE’s publication criteria as it currently stands. Therefore, we invite you to submit a revised version of the manuscript that addresses the points raised during the review process.

We look forward to receiving your revised manuscript.

Kind regards,

Carlos Alberto Antunes Viegas, DVM; MSc; PhD

Academic Editor

PLOS ONE

Journal Requirements:

Reviewers' comments:

Reviewer's Responses to Questions

**Comments to the Author**

1. If the authors have adequately addressed your comments raised in a previous round of review and you feel that this manuscript is now acceptable for publication, you may indicate that here to bypass the “Comments to the Author” section, enter your conflict of interest statement in the “Confidential to Editor” section, and submit your "Accept" recommendation.

Reviewer #3: (No Response)

2. Is the manuscript technically sound, and do the data support the conclusions?

Reviewer #3: Partly

3. Has the statistical analysis been performed appropriately and rigorously? 

Reviewer #3: Yes

4. Have the authors made all data underlying the findings in their manuscript fully available?

Reviewer #3: Yes

5. Is the manuscript presented in an intelligible fashion and written in standard English?

Reviewer #3: Yes

6. Review Comments to the Author

Reviewer #3: The limitations are listed. However, given the small sample size, the authors should put a qualification in the conclusions that, "All statistical inference (p-values especially) must be interpreted with caution, given the size of the sample.

7. PLOS authors have the option to publish the peer review history of their article (what does this mean?). If published, this will include your full peer review and any attached files.

Reviewer #3: No

---

## [Author Response · Author response to Decision Letter 3]

13 Feb 2024

Response to Reviewer #3: 

Comment: The limitations are listed. However, given the small sample size, the authors should put a qualification in the conclusions that, "All statistical inference (p-values especially) must be interpreted with caution, given the size of the sample.”

Answer: Thank you for your remarks. We added the sentence to the manuscript.

---

## [Decision Letter · Decision Letter 4]

20 Feb 2024

The effects of a synthetic epidermis spray on secondary intention wound healing in adult horses.

PONE-D-23-16094R4

Dear Dr. Charlotte Paindaveine,

We’re pleased to inform you that your manuscript has been judged scientifically suitable for publication and will be formally accepted for publication once it meets all outstanding technical requirements.

Kind regards,

Carlos Alberto Antunes Viegas, DVM; MSc; PhD

Academic Editor

PLOS ONE

Additional Editor Comments (optional):

Reviewers' comments:

Reviewer's Responses to Questions

**Comments to the Author**

1. If the authors have adequately addressed your comments raised in a previous round of review and you feel that this manuscript is now acceptable for publication, you may indicate that here to bypass the “Comments to the Author” section, enter your conflict of interest statement in the “Confidential to Editor” section, and submit your "Accept" recommendation.

Reviewer #3: All comments have been addressed

2. Is the manuscript technically sound, and do the data support the conclusions?

Reviewer #3: (No Response)

3. Has the statistical analysis been performed appropriately and rigorously? 

Reviewer #3: (No Response)

4. Have the authors made all data underlying the findings in their manuscript fully available?

Reviewer #3: (No Response)

5. Is the manuscript presented in an intelligible fashion and written in standard English?

Reviewer #3: (No Response)

6. Review Comments to the Author

Reviewer #3: (No Response)

7. PLOS authors have the option to publish the peer review history of their article (what does this mean?). If published, this will include your full peer review and any attached files.

Reviewer #3: No

---

## [Editor Report · Acceptance letter]

26 Feb 2024

PONE-D-23-16094R4 

PLOS ONE

Dear Dr. Charlotte C, 

I'm pleased to inform you that your manuscript has been deemed suitable for publication in PLOS ONE. Congratulations! Your manuscript is now being handed over to our production team.

Kind regards, 

on behalf of

Dr. Carlos Alberto Antunes Viegas 

Academic Editor

PLOS ONE